# Perspectives of Healthcare Professionals on Meaningful Activities for Persons with Dementia in Transition from Home to a Nursing Home: An Explorative Study

**DOI:** 10.3390/healthcare7030098

**Published:** 2019-08-19

**Authors:** Mari Groenendaal, Anne Loor, Manja Trouw, Wilco P. Achterberg, Monique A.A. Caljouw

**Affiliations:** 1WoonZorgcentra Haaglanden (Nursing Home Organization), the Hague, P.O. Box 9600, 2300 RC Leiden, The Netherlands; 2Department of Public Health and Primary Care, Leiden University Medical Center, 2300 RC Leiden, The Netherlands

**Keywords:** meaningful activities, transition, dementia, nursing home

## Abstract

Meaningful activities can enhance quality of life, a sense of connectedness, and personhood for persons with dementia. Healthcare professionals play an important role in maintaining meaningful activities, but little is currently known about the impact of the transition from home to nursing home on these activities. This study explored the experiences of professionals in four Dutch nursing homes, identifying facilitators and barriers to the maintenance of meaningful activities during the transition. A qualitative explorative design was used. Data were collected using focus groups and analyzed using thematic analysis. Twenty-two professionals participated in four focus groups, and three themes were identified: (1) a lack of awareness and attention for meaningful activities; (2) activities should be personalized and factors such as person characteristics, interests, the social and physical environment, and specific information such as roles, routines, activities, and personal issues play an important role in maintaining activities; (3) in the organization of care, a person-centered care vision, attitudes of professionals and interdisciplinary collaboration facilitate maintenance of meaningful activities. Healthcare professionals felt that meaningful activities are difficult to maintain and that improvements are needed. Our study provides suggestions on how to maintain meaningful activities for persons with dementia prior, during and after the transition.

## 1. Introduction

Meaningful activities are commonly mentioned in relation to quality of life and management of the behavioral and psychological symptoms of dementia [1]. The use of the term meaningful activity is widely spread, although in current literature, a clear definition of meaningful activity is lacking. In this study, meaningful activities are defined as all activities or occupations that are significant or meaningful for the person and reflect someone’s current and past interests, routines, habits, and roles and are adjusted to someone’s abilities [1,2]. A wide range of activities can be seen as meaningful activities, such as reminiscence, music, family, and social activities and individual activities related to routines and habits, for instance, domestic tasks [3]. For persons with dementia, meaningful activities can enhance wellbeing [4], provide a sense of connectedness to self, others, and the environment [5], and a sense of personhood [6]. Despite these benefits, engaging in meaningful activities can be difficult for persons with dementia, difficulties that are often related to the progress of dementia. One major and often inevitable challenge can be the transition to a nursing home. 

Moving to a nursing home is a major life event [7], and adjustment to a nursing home is difficult both for persons with dementia and their family members [8]. Most studies that have focused on the transition period and the factors influencing this transition period have described the perspectives of family members of persons with dementia [8,9] or focused on supporting family members [10]. A recent study investigated adjusting to residential aged care facilities from the perspectives of persons with dementia, family members, and facility care staff and identified meaningful activities as critical for facilitating adjustment to life in the facility [11]. A study by Rijnaard et al. [12] showed that engagement in activities and preservation of one’s habits and values in nursing homes can help persons with dementia to feel at home. 

Although the importance of meaningful activities for persons with dementia in nursing homes is known, little is known about factors influencing meaningful activities and maintenance of these activities in the transition period. One study of cognitively intact older persons showed a decline in meaningful activities when moving to a nursing home [13]. Tak et al. [14] have shown that while nursing home residents with dementia had a wide variety of hobbies and activities prior to moving to a nursing home, their current involvement in activities was limited and did not always match their interests. Further, the study of Davison et al. [11] found that although some residents continued with previous hobbies, many residents and families found activities inadequate to meet residents’ needs for stimulation and interest. In this study, staff found it difficult to find activities that meet individual interests, preferences, and capacities of persons with dementia and to incorporate meaningful activities in day-to-day-life. A range of factors are associated with activity involvement in the nursing home and relate to the person’s interests, competences, abilities, the (social and physical) environment, and other factors related to the organization of care [15]. Healthcare professionals play an important role in individually tailoring activities for persons with dementia by focusing on the individual’s preferences, interests, remaining abilities, memories or personal history [2]. This study explores the experiences of healthcare professionals in Dutch nursing homes in relation to (1) maintaining meaningful activities when persons with dementia move from home to a nursing home, (2) working together as a multidisciplinary team to maintain meaningful activities, and (3) the experienced barriers and facilitators.

## 2. Materials and Methods 

This study has a qualitative explorative design and used focus groups as a data collection method. Focus groups were used to gain an in-depth insight into the experiences of healthcare professionals regarding meaningful activities for persons with dementia in transition from the home to a nursing home, and the factors experienced as barriers and facilitators. Furthermore, focus groups were used because, in comparison with individual interviews, this method offers participants the opportunity to share their ideas and to interact and complement each other [16]. In this study, the transition period is seen as the period prior to, during, and the period following admission and therefore also includes the adjustment period [7]. 

### 2.1. Setting, Study Sample, and Recruitment

The participating care organization has nine nursing home locations in the region of the Hague, the Netherlands, which provide small-scale dementia living facilities within the environment of a larger nursing home. Six to eight persons live together in a group, with in total about 100 persons with dementia per nursing home location. Residents share a living room, and each has their own bedroom. Activities are integrated in daily care (e.g., cooking, household activities) or offered in groups (e.g., gym, dancing). Daily care is provided by a multidisciplinary team that consists of nurse aides, registered nurses, psychologists, elderly care physicians, occupational therapists, and physiotherapists. 

The inclusion criteria in this study required that the participating healthcare professionals were regular members of a multidisciplinary team and were employed in a dementia care unit. All teams were recruited by a researcher via email, through which a multidisciplinary team member in a dementia care unit was invited to explore the possibility if several multidisciplinary team members could participate in the focus group at their location. The member of the team who was approached by the researcher contacted the multidisciplinary team members also by email or face-to-face to see if they could participate in the focus group. Participation was on a voluntary basis.

### 2.2. Data Collection 

The focus groups were semistructured and had the following structure: (1) welcome, (2) focusing exercise, (3) main subject, and (4) completion. During the welcome, participants introduced themselves, and approval was obtained for audio recording. This was followed by a focusing exercise to introduce the topic of meaningful activity. The aim of this focusing exercise was to get the conversation going between the group members by introducing the topic [16]. Participants were asked to write examples of meaningful activities and what they believe makes an activity meaningful on Post-It notes, after which, discussion of their views took place. A different color of Post-It was used for each discipline in order to compare views per discipline. As the main subject of the focus group, various topics were discussed related to assessment and maintenance of meaningful activities during the transitional period. Opinions and experiences were shared concerning current practice, barriers and facilitators, and collaboration as a multidisciplinary team. The topic list for the main subject was derived from the experiences of the researcher (author 1) and an occupational therapist (author 3). The focusing exercise and main subject are listed in Table 1. During completion, all participants were able to contribute any last thoughts and ask questions about the study. 

All participants gave oral consent for audio recording of the focus groups and anonymous use of data. The focus groups were moderated by a researcher (author 1) and an occupational therapy student (author 2). The researcher is an experienced group facilitator, is employed in the organization as a lecturer practitioner, and has worked with multidisciplinary teams to maintain meaningful activities. Both moderators encouraged participation and discussion from all participants in the group interview. 

### 2.3. Data Processing and Analysis

The focus groups were analyzed by thematic analysis. Thematic analysis is a commonly used method to identify, analyze, and report patterns (themes) within data [17]. During the first step, familiarization with the data, author 2 listened to the audio recordings and transcribed the focus groups using the topic list. Author 1 then listened to the audio recordings and read the transcripts, comparing them to the audio recordings in an effort to identify any inaccuracies. The second step was to develop a coding frame, in which author 2 selected fragments from the transcripts and organized them based on the topic list. All the fragments were then read and coded by authors 1 and 2 and discussed based on the questions: ‘What is the fragment about?’ and ‘What does this fragment mean?’. The coding process was data-driven, meaning that the codes were based on the data, which resulted in a long list of specific codes [17]. The third step was to organize the codes into themes and subthemes; to facilitate this process, authors 1 and 2 used visual methods, such as tables, to structure their thinking process. In the fourth step, author 3 independently reviewed the themes, subthemes, and extracted data, reflecting on the coding process. Questions raised were discussed by authors 1, 2, and 3, and agreement was reached. The final step was to define and rename the themes. 

Authors 1 and 5 together reviewed the data, and three overall themes and several subthemes were identified and renamed. These themes were then verified by all other authors. Nvivo (version 12) was used for the coding and theme process, facilitating the structuring of the transcript into fragments, codes, subthemes, and themes. Anonymity and confidentiality were guaranteed, and the names of people, appearing in the quotes, were deleted from the written texts or were anonymized. A member check was conducted on the findings; the participants found the research findings reflecting their meanings and perceptions [18].

## 3. Results

Four nursing homes indicated an interest in participating in the study. The remaining five declined to participate due to a shortage of staff related to the vacation season. In total, 22 healthcare professionals participated in four focus groups. The focus groups had a broad composition and included a range of professionals comparable to the typical multidisciplinary team in a Dutch nursing home, represented in the following disciplines: five psychologists, four physiotherapists, three nurse aides, one registered nurse, one hostess (a paid assistant without nurse training), one activity therapist, three occupational therapists, two team managers, one physician, and one elderly care physician. The number of participants in the four focus groups varied between three and seven: Two focus groups had six participants, one had seven, and one had three. In total, 4 men and 18 women participated. Work experiences in the care for persons with dementia varied: Four participants had 0–2 years’ experience, nine participants 2–10 years, and seven participants more than 10 years, while we had no data from two participants. Six participants were 20–35 years of age, nine participants 35–50, three participants 50–65, and we had no data from four participants. The focus groups convened from 10 July to 24 July 2018. Two of the focus groups required half an hour, and two required one hour. 

Three themes were identified in the data which, according to the healthcare professionals, influenced the maintenance of meaningful activities as persons with dementia transitioned from home to a nursing home. These themes were (1) attention and awareness, (2) personalized meaningful activities, and (3) the organization of care.

### 3.1. Attention and Awareness

During the focusing exercise, healthcare professionals were asked to give examples of meaningful activities for persons with dementia. Five of the eight disciplines mentioned “moving and/or walking”, “music”, and “kitchen activity and/or cooking” as meaningful activities. A wide variety of examples were given, both between and within the disciplines, varying from leisure activities to activities related to routines, roles and habits. Not every activity was considered especially meaningful by all disciplines.

Participants were also asked which factors gave meaning to an activity. The items “based on interests” and “mood enhancing” were mentioned by six disciplines and “adapted from past activities” and “personalized activities” by five disciplines. In each focus group, all attendees could agree with the items mentioned by a different discipline.

Healthcare professionals in all focus groups reported that meaningful activities receive little attention and constitute a minor aspect of the transition period, except when this presents a problem or leads to behavioral problems in the person with dementia. Nevertheless, they believe that greater attention for this subject could be beneficial.
“We can definitely improve care for persons who become distressed in their new environment. For example, if a person appears to have behavioral problems or aggression following admission, we need to identify factors appropriate to that person”*(Team manager, focus group 4)*

In addition, the professionals argued that persons without behavioral problems also deserve attention, because these individuals may be at risk “of quietly disappearing into the crowd or of being overlooked”. In general, professionals endorsed the value of meaningful activities in increasing quality of life and providing a sense of identity and meaning.
“But it is also important to understand the patterns of someone’s home life before admission and how we can help that individual to maintain that life in the nursing home. This will necessarily differ because a nursing home is not someone’s original home, but we should try to identify useful elements. So that someone still feels “I matter”, “I’m involved” and “I’m important”.”*(Psychologist, focus group 3)*

Healthcare professionals report a lack of awareness amongst colleagues and family members regarding meaningful activities. For family members, it can be difficult to recognize which activities could be meaningful and how these activities can be maintained, an issue probably related to the fact that family members have experienced the decline and loss of activities firsthand and have consequently come to believe that the person with dementia cannot regain activities. Another issue is that some family members consider that older persons no longer need to undertake activities. A final reason family members may be reluctant to share in activities and routines could be embarrassment related to the behavior of the person with dementia. The participating professionals proposed that additional information could be helpful for family members. In addition to creating awareness among family members, improved awareness among healthcare professionals and giving priority to meaningful activities were considered important, especially in the transitional period characterized by other urgent considerations (e.g., medical problems and issues).
“There is insufficient awareness and professionals ask too few questions related to meaningful activities. In essence, there is often too little awareness of its importance. The question is why someone is here; and that is not because they once had nice hobbies. The medical condition receives primary attention.”*(Psychologist, focus group 2)*

Healthcare professionals suggest that more awareness and attention could be created by integrating meaningful activities into standard care. During these procedures, it is important to personalize meaningful activities, a theme elaborated below.

### 3.2. Personalizing Meaningful Activity

Healthcare professionals stated that both general and personalized activities are available in nursing homes. General activities are offered in groups and have a specific program or are performed in the living room. These activities are usually developed for and are intended to meet the needs of residents but are not personalized. Although these general group activities are valuable, the participants suggested that a combination of personalized and (group) activities would be optimal.
“I think there should be a combination. People can sometimes no longer properly express their wishes, or family members find it difficult to be specific. Sometimes it is good to offer general activities for people to enjoy, but that is not enough; it is also important to define what someone wants and needs.”*(Occupational therapist, focus group 3)*

In general, professionals felt that activities decline after moving to a nursing home. Nonetheless, some people become more active following admission due to clear structure in their day and greater stimuli (see theme three). A range of factors were mentioned as influencing the performance and maintenance of meaningful activities in the transition from home to a nursing home. Factors mentioned included the cognitive and physical abilities of the person with dementia, a change in interests, and the changing environment (both in terms of the building itself and social factors). As a consequence of the decline in the cognitive and physical abilities of persons with dementia, professionals thought it important to adjust activities to meet abilities. Persons with dementia can experience frustration and loss of motivation when offering them activities that they previously performed. In addition, interest and motivation can change as a result of the disease or the change in environment. Activities previously performed at home may no longer be meaningful after transition to a nursing home.
“Certain activities can be sustained, but in my experience if someone no longer finds an activity meaningful they will not want to do it anymore, such as peeling potatoes, for example.”*(Nursing aide, focus group 3)*

Healthcare professionals often experience barriers to maintaining meaningful activities related to a lack of materials and to personal issues stemming from the move, reduced accessibility to and from the unit, and a different room layout and outside area.
“…for example, when someone is used to a small living area and has everything close to hand. This is completely different in a nursing home with a long corridor leading to the living room”.*(Physiotherapist, focus group 4)*

A facilitator of a successful transition is the possibility of continuing socially meaningful activities, for example, birthdays and holidays, and the involvement of family members is considered important in preserving someone’s living patterns. 

Healthcare professionals argued that specific changes are needed to maintain meaningful activities. They state that understanding the reason someone performed a given activity can be important to find an appropriate alternative activity adjusted in a way that is meaningful.
“What did the activity represent and what did it mean to someone. For example, going outside gave a sense of freedom, so how can you now encapsulate this sense of freedom in a still achievable activity”.*(Psychologist, focus group 2)*

Another facilitator supporting the maintenance of meaningful activities is finding meaning in ‘little things’, such as the daily routines someone was used to before moving to a nursing home. Specific details of a person’s life history are collected during the period immediately following admission, and these can help in the search for possible activities, although additional information is often needed. The healthcare professionals stated that information on the final period at home is especially important.
“At admission we ask questions about hobbies. But you should also look more specifically at a person’s role at home and the activities they undertook when still at home”.*(Physiotherapist, focus group 3)*

The last facilitator to be mentioned was the need for reflection on the implementation of an activity in daily nursing home practice. If the activity is easy to implement, it is more likely that the activity will take place. The focus groups mentioned additional organizational issues, which are discussed in the next theme.

### 3.3. Organization of Care

This theme includes several subthemes, with facilitators and barriers defined based on the data from the focus groups: (1) person-centered care vision and attitudes of professionals, (2) interdisciplinary collaboration, and (3) requirements (See Table 2). 

In three focus groups, the participants felt that a person-centered focus in the attitude of professionals and organization of care in the nursing home is needed in order to realize meaningful activities. Attitudes focused on the questions a person may have, including asking a person what is important to them, contribute to personalized meaningful activities. Recently, a more person-centered approach was implemented in the multidisciplinary team meetings in the care organization. Following this change, the professionals noticed that more questions were being asked in the multidisciplinary team meeting related to meaningful activities. Although the overall opinion was that meaningful activities deserve greater attention, the multidisciplinary team meetings were mentioned in all focus groups as a facilitator of this idea. In general, an organization in which professionals seek opportunities, are flexible, and apply fresh thinking will be supportive of maintaining meaningful personalized activities. 

By contrast, in three focus groups, the participants argued that adopting tasks and structures and routines from the perspective of professionals and the organization could represent a threat to the maintenance of meaningful activities.
“When someone enters a nursing home, it is often said that they must first ‘get used to’ the new environment and ‘settle down’ for a couple of weeks. Actually, you are allowing someone to become hospitalized during those two weeks. There is a danger in doing nothing for too long, with the result that routines and habits are lost.”*(Psychologist, focus group 1)*

Although to facilitate person-centered care, a flexible approach to work was considered important, also, a single team approach was seen as a facilitator by two focus groups. This issue is further elaborated in the subtheme of interdisciplinary collaboration. 

Many different healthcare professionals are involved in the assessment and realization of meaningful activities. For example, information is gathered during planned conversations with family members and with the person with dementia, but also during care moments and conversations throughout the day. Two focus groups felt that when this information is not integrated, it can act as a barrier to the maintenance of meaningful activities. Several situations were mentioned whereby this information could be aligned, both formal and informal. One was related to the multidisciplinary team meeting and another to the possibility of working together as a single team in the nursing home. No contact outside of formal meetings was mentioned as a barrier.
“The lines of communication between all disciplines are short (hostess). We really feel it is one team (psychologist). And because the lines are short, professionals are also prepared to ask more of each other (physiotherapist).”*(Focus group 4)*

Goal setting can facilitate the process of finding alignment. Although the finding and formulation of goals can be complicated, the overall opinion was that they are important in realizing meaningful activities and are helpful in making a plan specific and explicit.
“The plan is often unclear, as are the conditions which must be met. For example, omeone likes Elvis’s music, but can they put on the music themselves, is equipment available, who is responsible for it and how does someone react to the music.”*(Psychologist, focus group 1)*

In all focus groups, clear responsibilities were considered important due to the many different disciplines involved, so working together as a team was viewed as essential. It is important to know who is responsible for what and to undertake regular evaluations of agreements.
“Available care is developed around the residents. Each discipline carries out its own assessment and we jointly discuss how they can be combined and implemented, resulting in a cooperative, dynamic process. There are no clear boundaries as to who should do what, as the focus is on the integrated package (nurse). We are integrating more of our work (psychologist). The ball keeps rolling because everyone cooperates and feels jointly responsible” (nurse).”*(All focus group 4)*

A factor experienced as a barrier was the lack of input from professionals involved prior to admission to the nursing home, for example, the care professionals at the daycare center or those providing home care. One focus group mentioned as a facilitator the possibility for a person with dementia to acclimatize to their new situation, for example, by visiting the nursing home before moving in.

Various requirements are mentioned in the maintenance of meaningful activities. The facilitators were access to care plan for all involved in care, the presence of a hostess, and a permanent team in the nursing home. The barriers to this included time, shortage of staff, and temporary workers.

## 4. Discussion

This qualitative, explorative study identified three themes important to the maintenance of meaningful activities for persons with dementia during the transition from home to a nursing home. The first theme was awareness and attention for meaningful activities during the period of transition on the part of healthcare professionals and family members. The second theme involved how to adapt and personalize an activity to a new environment and to adapt the activity so that it remains meaningful. The participating healthcare professionals argued that specific information is needed on the involvement of family members and the activities, roles, and routines someone preformed in the period before moving to a nursing home. In addition, various factors that may influence the performance of activities have to be taken into account (person characteristics, activity, and environment). The third and last theme embodied what is needed in the organization of care in order to maintain meaningful activities.

The healthcare professionals consulted in this study reported that meaningful activities are difficult to maintain during the transition from home to a nursing home and are maintained less often than desired. These conclusions correspond to the findings of an earlier study which reported that residents of nursing homes are generally inactive and consequently have low overall activity levels [19]. Currently, no studies specifically describe investigations of how to adjust activities to remain meaningful in the transition from home to a nursing home. There are, however, studies that focus on how to moderate activities. For instance, Regier et al. [20] described a model for caregivers illustrating how to design activities that maximize engagement. In this model, the remaining cognitive and physical functional abilities are aligned to interests and environmental characteristics. These factors are similar to those found in our study, in which healthcare professionals mentioned that various factors are important in the maintenance of meaningful activities related to the remaining cognitive and physical abilities of the person with dementia, the possible change in interests, and the changing environment (both the building and social) due to moving. In addition, the motivation underlying an activity and a person’s life history are important to the personalization of a meaningful activity and to adaptations of an activity to the new environment. Du Toit, Shen, and McGrath [21] also stressed the importance of a deep understanding of why activities were undertaken in order to promote person-centered care. Viewing person-centered care as the foundation can help in finding an alternative activity that matches the values or needs embedded in the previous activity. Especially as dementia progresses, an individual may no longer be able to engage in their valued meaningful activity, even when using compensation strategies or adaptive tools or equipment [5]. In our study, we found that specifically assessing someone’s activities before admission to a nursing home, determining the underlying meaning of activities for the person with dementia, and involving family members in any assessment may all contribute to identifying and maintaining meaningful activities. In this study, healthcare professionals experienced a lack of attention and awareness among healthcare professionals within their nursing home organizations, which could have an influence on the identification and maintenance of meaningful activities in transition from home to a nursing home. The participating professionals expressed the view that a transition in living environment could have an impact on maintaining activities for persons with dementia. Although a more structured environment and daily care might be beneficial for some individuals, most people risk losing activities. There is little research at present on the impact of a transition from home to a nursing home on meaningful activities. The findings reported in the present study suggest that adaptations will be required in order to ensure that personalized meaningful activities remain meaningful in a new environment.

Interdisciplinary collaboration, with healthcare professionals in the nursing home and professionals involved prior to the admission, was found to be a facilitator in the organization of care, permitting the realization of meaningful activities for persons with dementia in the transition from home to a nursing home. In a systematic review [22], interdisciplinary interventions were found to have a positive effect on patient outcomes in nursing homes. To improve transitional care focused on maintaining meaningful activities, an interdisciplinary approach, both with professionals in the nursing home and professionals involved prior to the admission, is therefore recommended.

Strengths of this study included the participation of a variety of different disciplines in the focus groups, which led to a better understanding of the facilitators and barriers healthcare professionals experience in daily practice. The focus groups consisted of persons working together on a daily basis, which allowed for an in-depth discussion about their own situation and work process. Professionals in the organization that provided the focus groups were familiar with meaningful activities, and a person-centered care approach has recently been implemented. This allowed the professionals to draw on their own experience in the maintenance of meaningful activities and to reflect on and formulate facilitators and barriers. A limitation of the study was that it was conducted in four nursing homes allied to the same care organization, a factor that may conceivably have influenced the diversity of the information provided by the healthcare professionals. Another limitation was that author one, who moderated the focus groups, also worked in the organization. Although we cannot rule out that this had an effect on the participants’ responses in the focus groups [18], we do not have any indication for different responses in the focus groups because of the affiliation of the researchers. Further, the cooperation with researchers that are well experienced in Dutch nursing home care but outside this organization reduced the risk of overly biased results.

## 5. Conclusions and Recommendations for Practice and Research 

To the best of our knowledge, this is the first study which explored the experiences of healthcare professionals regarding the maintenance of meaningful activities for persons with dementia during the transition from home to a nursing home. Providing personal meaningful activities for persons with dementia is a challenge for healthcare professionals and healthcare organizations [11,14,19]. Furthermore, there is a lack of guidelines [11] and interventions [8,10] focusing on the transition from home to a nursing home for persons with dementia. The findings of this study provide valuable insights, both for professionals and organizations, on how to maintain meaningful activities during this transition. Several recommendations could be made for healthcare professionals and organizations based on the findings in the present study. Training of healthcare professionals can contribute to awareness of the importance of meaningful activities in the transition period, knowledge of the factors influencing the maintenance, risks of losing meaningful activities, and possibilities to maintain meaningful activities. The results of the study highlight the need for the development of an interdisciplinary multicomponent intervention and guidelines aimed at supporting persons with dementia and their families during this important transitional period. Further research should focus on how interventions could help persons with dementia and family members to maintain their lives as fully as possible during the transitional period and how activities can be adapted to remain meaningful for them, despite progression of the disease and changing environments.

## Figures and Tables

**Table 1 healthcare-07-00098-t001:** Focusing exercise and main subject.

Head Topic	Sub Topic
What is a meaningful activity?	Can you give an example of a meaningful activity? What do you believe makes an activity meaningful?
How are meaningful activities assessed?	Opinions and current practice: What do you consider barriers and facilitators? How do you collaborate as a multidisciplinary team in the assessment of meaningful activities?
How do you maintain meaningful activities during the transition from home to nursing home?	Opinions and current practice: What do you consider barriers and facilitators? How do you collaborate as a multidisciplinary team in maintaining meaningful activities?

**Table 2 healthcare-07-00098-t002:** Facilitators and barriers to maintaining meaningful activities in the transition period as experienced by healthcare professionals.

Facilitator	Barrier
**Theme 1: AWARENESS**	
Multidisciplinary team meetings promote greater awareness by professionals by asking questions such as: What is meaningful for this person? What are the things this person wants to do?	A lack of awareness of meaningful activities among professionals
Greater attention in standard procedures for meaningful activities during the transition period	A lack of awareness by family members: little belief that activities can be regained, and a view that older persons do not need to maintain activities
**Theme 2: PERSONALIZED MEANINGFUL ACTIVITY**	
Questions such as: What makes you happy? “What is important for a day to be a good day?”	Insufficient specific knowledge of activities, roles and habits before moving to the nursing home
Both general and personalized activities are offered	Only general activities available, rather than asking which activities are preferred
Family involvement	Lack of specific information about activities from family members
Environment provides the possibility to perform activities	Environment: accessibility of the unit, change of environment (moving) can change routines of daily living. Materials unavailable and personal issues related to moving. Change of room layout
Determining why an activity is performed, finding satisfaction in small things and the opportunities for implementing the activity in daily routines and nursing home daily practice	Factors relating to the person with dementia: advanced dementia, apathy, frustration
**Theme 3: ORGANISATION OF CARE**	
*Subtheme person-centered care vision and attitudes of professionals*	
Professionals seek opportunities, are flexible, apply fresh thinking, stimulating persons with dementia and experimenting	Adopting tasks and care
Structuring the day, a single team approach	Holding on to structures and routines from the perspective of the organization
Multidisciplinary team meeting, focusing on meaningful activities and well-being	
*Subtheme: Interdisciplinary collaboration*	
Goal setting, paying attention to specific conditions	Non-specific goals/information not aligned
Interdisciplinary collaboration: no specific division of roles	Few exchanges between professionals
Presence of professionals at the care location. An involved manager who thinks in possibilities	Few possibilities for exchanges except in multidisciplinary team meetings
Possibility of acclimatizing to the new environment	No exchanges with previously involved professionals from other organizations
*Subtheme: requirements*	
Care plan access for all involved in care Presence of hostess Permanent team	Time, shortage of staff Temporary workers

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
