# Peer review of "Perspectives of Healthcare Professionals on Meaningful Activities for Persons with Dementia in Transition from Home to a Nursing Home: An Explorative Study"

_healthcare, 2019, doi:10.3390/healthcare7030098_

Round 1

Reviewer 1 Report

The quality of the manuscript has improved based on the changed made by the authors.

By adding definitions of concepts such as 'transition' and 'meaningful activities' it the manuscript has become more clear. 

Furthermore, clarifications on methodological issues such as the recruitment, and member checking were added.

The added value of the current manuscript is more evident based on the changes in the discussion.

I have no additional comments.

This manuscript is a resubmission of an earlier submission. The following is a list of the peer review reports and author responses from that submission.

Round 1

Reviewer 1 Report

The authors are writing about an interesting and important topic. Facilitating participation in meaningful activities in a nursing home remains a challenge and more research on how to do this is recommended. 

I do have some points that require clarification in my opinion.

General:

- In the whole manuscript the authors describe that the aim of the study is to gain insight into the experiences of healthcare professionals regarding meaningful activities for persons with dementia in transition from home to a nursing home. However I believe your study focuses more on the period after the transition (transition can be divided into prior, during and after the transition). 

- There are some language/typing mistakes, for example:

    - 44: 'cognitively older persons' should be 'cognitively impaired older persons', or at least something like that..

    - 213: 'in a way than' should be 'in a way that'

    - also, some sentenced are difficult to understand, please check the manuscript for these.

Introduction:

- The introduction starts with a paragraph on meaningful activities, however a definition (and examples) is lacking. It would be good to add this for clarification.

- Part of the introduction is about the transition process, whereas the focus of the study should be more on the adjustment period after the transition (and the difficulties faced there).

Methods:

- Design: I believe you have used a qualitative exploratory design and used focus group as a data collection method (so no explorative, qualitative focus group design)

- The recruitment should be clarified. Is one member of a team participating, or a whole team (74-77).

- 94 - focus groups where moderated, not facilitated..

- Some methodological considerations should be explained further (I advise to check the Qorec checklist):

    - Was a member check conducted? if not, this should be explained... perhaps as a limitation.

    - How did the authors control for possible bias?

    - Data saturation?

    - where other characteristics also gathered (age, gender, years of working experience)? if not,     this could be a limitation as well..

Results:

The overall themes are explained clearly. I would have expected more information about the differences between disciplines (as in the methods it is described that different colors of post-its were used etc.)

284 - what is the difference between a precondition and a facilitator?

Discussion:

I believe some limitations could have been added (based on my comments in the methods).

Furthermore: was there any dependability between respondents? in such a way that it could have influenced their input?

What is the paper adding should be emphasized as well, as the findings do not seem surprising compared with previous studies.

Author Response

Reviewer 1

The authors are writing about an interesting and important topic. Facilitating participation in meaningful activities in a nursing home remains a challenge and more research on how to do this is recommended.

Thank you for your compliments and your well-considered comments. We hope we sufficiently clarify your points in the answers and article.

Q1- In the whole manuscript the authors describe that the aim of the study is to gain insight into the experiences of healthcare professionals regarding meaningful activities for persons with dementia in transition from home to a nursing home. However I believe your study focuses more on the period after the transition (transition can be divided into prior, during and after the transition).

A1 Although all participants are involved in care in the nursing home they reflect on the whole period of the transition (prior, during and after). But we can enhance with the reviewer’s comment that there is more evidence in the article on the period after the admission instead of the period prior to and during. For clarification, we added a definition regarding transition as used in this study in line 81 page 2 of the article. This definition reads: “In this study, the transition period is seen as the period prior to, during and the period following admission and therefore also includes the adjustment period”.

Q2- There are some language/typing mistakes, for example:

    - 44: 'cognitively older persons' should be 'cognitively impaired older persons', or at least something like that..

    - 213: 'in a way than' should be 'in a way that'

    - also, some sentenced are difficult to understand, please check the manuscript for these.

A2: Thank you for your precise reading.

The word ‘intact’ disappeared in the final version of the article. This was a typographical error. We changed the sentence and it reads now: ‘One study of cognitively intact older persons….’ (line 57 page 2).

Also, we changed ‘in a way than’ into ‘in a way that’ (line 246 page 6)

Introduction

Q3.1:- The introduction starts with a paragraph on meaningful activities, however a definition (and examples) is lacking. It would be good to add this for clarification.

A3.1: We added a definition in the article (line 32 page 1). The use of the term meaningful activity is widely spread, although in current literature a clear definition of meaningful activity is lacking. In this study, meaningful activities are defined as all activities or occupations that are significant or meaningful for the person and reflect someone’s current and past interests, routines, habits and roles and are adjusted to someone’s abilities [1,2]. A wide range of activities can be seen as meaningful activities such as reminiscence, music, family and social activities and individual activities related to routines and habits, for instance domestic tasks [3].

Q3.2 Part of the introduction is about the transition process, whereas the focus of the study should be more on the adjustment period after the transition (and the difficulties faced there).

A3.2 Recently a study has been published about the adjustment period and the perspectives of persons with dementia, family members and facility staff [4]. We have added the results of this study in the introduction (line 46 page 2): A recent study investigated adjusting to residential aged care facilities from the perspectives of persons with dementia, family members and facility care staff and identified meaningful activities as critical for facilitating adjustment to life in the facility. And in line 61 page 2:  Also the study of Davison et al.[4] found that although some residents continued with previous hobbies, many residents and families found activities inadequate to meet residents’ needs for stimulation and interest. In this study staff found it difficult to find activities that meets individual interests, preferences and capacities of persons with dementia and to incorporate meaningful activities in day-to-day-life.

Methods:

Q4.1 Design: I believe you have used a qualitative exploratory design and used focus group as a data collection method (so no explorative, qualitative focus group design)

A4.1 We changed this according to your suggestion in line 75 page 2. The new sentence now reads: ‘This study has  a qualitative explorative design and used focus groups as a data collection method.’

Q4.2 The recruitment should be clarified. Is one member of a team participating, or a whole team (74-77).

A4.3 The researcher approached one member of the multidisciplinary team in all nine nursing homes of the organization by email. In this email, the goal of the focus group was explained, as well as the duration and inclusion criteria. This member was also asked if they could ask whether several members multidisciplinary team members could participate in the focus group at their location. The member of the team who was approached by the researcher, contacted the multidisciplinary team members also by email or face-to-face to see if they could participate in the focus group.

We’ve clarified this process in more detail in the article (line 95, page 3).

Q4.3 - 94 - focus groups where moderated, not facilitated..

A4.3 Thank you for this suggestion we’ve replaced facilitated by moderated  (line 118 and 121 page 3)

Q4.4 Some methodological considerations should be explained further (I advise to check the Qorec checklist):

A4.4 Thank you for suggesting to check the COREQ checklist.

Q4.5 Was a member check conducted? if not, this should be explained... perhaps as a limitation.

A4.5 The study outcomes were presented in several presentations in the organization, also some participants received the results section of the article to check if it reflected their opinions. Participants with whom the findings were discussed, found that the findings reflected their own meanings and perspectives. We added this in the method section line 144 page 4 as followed: A member check was conducted on the findings, the participants found the research findings reflecting their meanings and perceptions [5].

Q4.6 How did the authors control for possible bias?

A4.6 As this is a qualitative study, results have been subjected to personal attitudes and experiences of the researchers. As written in line 119 page 3 authors one, who moderated the focus groups also worked in the organization. Although we cannot rule out that this had an effect on the participants responses in the focus groups [5], we do not have any indication for different responses in the focus groups because of the affiliation of the researchers. Next to the member check we discussed above, the cooperation with researchers that are well experienced in Dutch nursing home care, but outside this organization reduced the risk of overly biased results. We added this as followed in the discussion (line 384 page 10): Another limitation was that author one, who moderated the focus groups also worked in the organization. Although we cannot rule out that this had an effect on the participants responses in the focus groups [5], we do not have any indication for different responses in the focus groups because of the affiliation of the researchers. Besides, the cooperation with researchers that are well experienced in Dutch nursing home care, but outside this organization reduced the risk of overly biased results.  

Q4.7 Data saturation?

A4.7 Data saturation was not discussed in the article. We conducted four focus groups, which were planned beforehand. Data saturation was reached at the end of the last focus groups, because no new major themes emerged in the data.

Q4.8 where other characteristics also gathered (age, gender, years of working experience)? if not,     this could be a limitation as well.

A4.8 We obtained characteristics about the participants, see the table 1 below and added a summary of the characteristics in the article line 155 page 4. In total four men and 18 women participated. Work experiences in the care for persons with dementia varied: four participants had 0-2  years’ experience, nine participants 2-10 years and seven participants more than 10 years, from two participants we had no data. Six participants were between 20-35 years of age, nine participants between 35-50, three participants between 50-65, from four participants we had no data.

Results:

Q5.1 The overall themes are explained clearly. I would have expected more information about the differences between disciplines (as in the methods it is described that different colors of post-its were used etc.)

A5.1 Although we used different colors of post-it’s to distinguish between the disciplines, we did not found, as expected beforehand, differences between their experiences. 

Results:

Q5.2 284 - what is the difference between a precondition and a facilitator?

A5.2 To clarify, the term ‘precondition’ has been replaced with ‘requirement’ in the article (line 265 page 6 and line 317 page 7). Requirement is a necessary condition, in this case, to maintain meaningful activities. Whereas facilitators and barriers are factors that either enable or counteract that requirement.

Discussion:

Q6.1 I believe some limitations could have been added (based on my comments in the methods).

A6.1 we added a limitation in the discussion session based on the comments (line 384 page 10): Authors one, who moderated the focus groups also worked in the organization.  Although this could have had an effect on the participants responses in the focus groups [5], we did not notice any of this during the focus groups. Besides, the cooperation with researchers that are well experienced in Dutch nursing home care, but outside this organization reduced the risk of overly biased results.  

Q6.2 Furthermore: was there any dependability between respondents? in such a way that it could have influenced their input?

A6.2 In two focus groups, there was a team manager present, although this could have been of influence, we did not notice any dependability in these focus groups compared to the focus groups without a team manager present. In all groups, the participants shared positive and negative experiences.

Q6.3 What is the paper adding should be emphasized as well, as the findings do not seem surprising compared with previous studies.

A6.4 What this paper adds:

  • This study shows that health care professionals experienced a lack of attention and awareness among health care professionals within their nursing home organizations which could have an influence on the identification and maintenance of meaningful activities in transition from home to a nursing home (added in article in line 358 till 361 page 9).
  • Paying specific attention to collecting information about meaningful activities before and after the move to the nursing home and adapting meaningful activities to the new situation can contribute to the maintenance of meaningful activities (in the article line 355 page 9 , the word maintaining was added)
  • Improving transitional care focused on maintaining meaningful activities, an interdisciplinary approach, both with professionals in the nursing home and professionals involved prior to the admission, is therefore recommended. (in article line 372 page 10)

To our best knowledge, this explorative study is the first study that focus completely on maintaining meaningful activities for persons with dementia in the transition period from home to a nursing home from the perspective of health care professionals.  (in article line 391 page 10)

In the search for healthcare adjusted to the need of older people and to support older people to live the life they wish to live, person centred care, the finding of this study provides valuable insights. (last part of this sentence in article line 397 page 10).  Especially because the transition is experienced as a major life event for persons with dementia and their family members [6] and adjustment is difficult [7]. Meaningful activities are mentioned as essential for adjusting to residential care [4].  A range of factors are associated with activity involvement in the nursing home and relate to the person’s interests, competences, abilities, the (social and physical) environment and other factors related to the organization of care [8].  (above sentences are present in the introduction) Providing personal meaningful activities for persons with dementia is a challenge for health care professionals and health care organisations [7,9,10]. Furthermore, there is a lack of guidelines [4] and interventions [7,11] focusing on the transition from home to a nursing home. (sentences are added in line 393 page 10). The findings of this study provide valuable insights, both for professionals and organizations, on how to maintain meaningful activities during this transition. (in article line 396 page 10)

Table 1. Participants demographics

Focus group

n

occupation

gender

Work experience in the care for persons with dementia

age

1.

6

Occupational therapist (n=2), Activity therapist (n=1),Team manager (n=1), Nurse aides (n=1), Psychologist (n=1)

Five women

one men

2-10 years: n=1

>10 years: n=3

No data: n=2

35-50 year: n= 3

50-65 years: n=1

No data: n=2

2.

3

Physician(n=1),Elderly care physician (n=1), Psychologist(n=1)

One women

Two men

0-2 years: n=1

2-10 years: n=1

>10 years: n=1

20-35: n=1

35-50 year: N=2

3.

6

Occupational therapist(n=1), Psychologist(n=2), nurse aides(n=1), Physiotherapist(n=2)

Six women

0-2 years: n=1

2-10 years: n=4

>10 years: n=1

20-35: n=3

35-50 year: N=2

No data: n=1

4.

7

Psychologist (n=1), nurse aide(n=1), registered nurse(n=1), hostess(n=1), team manager(n=1), physiotherapist(n=2)

Six women

One men

0-2 years: n=2

2-10 years: n=3

>10 years: n=2

20-35: n=2

35-50 year: N=2

50-65 years: n=2 No data: n=1

total

N=22

five psychologists, four physiotherapists, three nurse aides, one registered nurse, one hostess (a paid assistant without nurse training), one activity therapist, three occupational therapists, two team managers, one physician and one elderly care physician.

18 women

4 men

0-2 years: n=4

2-10 years: n=9

>10 years: n=7

No data: n=2

20-35: n=6

35-50 year: n= 9

50-65 years: n=3

No data: n=4

References

  1. Travers, C.; Brooks, D.; Hines, S.; O'Reilly, M.; McMaster, M.; He, W.; MacAndrew, M.; Fielding, E.; Karlsson, L.; Beattie, E. Effectiveness of meaningful occupation interventions for people living with dementia in residential aged care: a systematic review. JBI Database System Rev Implement Rep 2016, 14, 163-225, doi:10.11124/jbisrir-2016-003230.
  2. Han, A.; Radel, J.; McDowd, J.M.; Sabata, D. The Benefits of Individualized Leisure and Social Activity Interventions for People with Dementia: A Systematic Review. Activities, Adaptation & Aging 2016, 40, 219-265, doi:10.1080/01924788.2016.1199516.
  3. Harmer, B.J.; Orrell, M. What is meaningful activity for people with dementia living in care homes? A comparison of the views of older people with dementia, staff and family carers. Aging Ment Health 2008, 12, 548-558, doi:10.1080/13607860802343019.
  4. Davison, T.E.; Camões Costa, V.; Clark, A. Adjusting to life in a residential aged care facility: Perspectives of people with dementia, family members and facility care staff. Journal of Clinical Nursing 2019, 10.1111/jocn.14978, doi:10.1111/jocn.14978.
  5. Tong, A.; Sainsbury, P.; Craig, J. Consolidated criteria for reporting qualitative research (COREQ): a 32-item checklist for interviews and focus groups. Int J Qual Health Care 2007, 19, 349-357, doi:10.1093/intqhc/mzm042.
  6. Afram, B.; Verbeek, H.; Bleijlevens, M.H.; Hamers, J.P. Needs of informal caregivers during transition from home towards institutional care in dementia: a systematic review of qualitative studies. Int Psychogeriatr 2015, 27, 891-902, doi:10.1017/S1041610214002154.
  7. Sury, L.; Burns, K.; Brodaty, H. Moving in: adjustment of people living with dementia going into a nursing home and their families. Int Psychogeriatr 2013, 25, 867-876, doi:10.1017/S1041610213000057.
  8. Smit, D.; de Lange, J.; Willemse, B.; Pot, A.M. Predictors of activity involvement in dementia care homes: a cross-sectional study. BMC Geriatr 2017, 17, 175, doi:10.1186/s12877-017-0564-7.
  9. Tak, S.H.; Kedia, S.; Tongumpun, T.M.; Hong, S.H. Activity Engagement: Perspectives from Nursing Home Residents with Dementia. Educ Gerontol 2015, 41, 182-192, doi:10.1080/03601277.2014.937217.
  10. den Ouden, M.; Bleijlevens, M.H.; Meijers, J.M.; Zwakhalen, S.M.; Braun, S.M.; Tan, F.E.; Hamers, J.P. Daily (In)Activities of Nursing Home Residents in Their Wards: An Observation Study. J Am Med Dir Assoc 2015, 16, 963-968, doi:10.1016/j.jamda.2015.05.016.
  11. Muller, C.; Lautenschlager, S.; Meyer, G.; Stephan, A. Interventions to support people with dementia and their caregivers during the transition from home care to nursing home care: A systematic review. Int J Nurs Stud 2017, 71, 139-152, doi:10.1016/j.ijnurstu.2017.03.013.

Reviewer 2 Report

This is an interesting manuscript that addresses an area of importance. There are several factors that need to be addressed.

1)    Please can the authors provide an account of what ethical approvals were sought and how they gained the necessary permissions to access the healthcare professionals. How as consent obtained? What information was provided to potential participants. How did the researchers ensure anonymity and confidentiality? Where were the focus groups conducted? What was the role of the Occupational Therapy student in the investigation and who invited them to join the research team?

2)    Recruitment of the participants was via e-mail (page 2 line –75) how did the researchers obtain this personal data and gain access to this information? What information was sent via e-mail? Did the researchers work in the participating institutions?

3)    Page 2 line 89 – 90  “The topic list for the main subject was derived from the experiences of the researcher (author 1) and an occupational therapist (author 3). What did the topic list comprise of? What is the experience of the researcher?

4)    Can an overview of the 22 participants be presented – this is briefly outlined on page 4 lines 121 -129. Was any further demographic information collected – if so, can a summary be provided? It would be useful to provide participant information for each of the focus groups.

5)    The term fragment is used page 3 lines 100 -117 – please can you clarify are you referring to extracts/excerpts of the data transcripts – also who undertook the transcription of the focus group interviews?

6)    Please can the authors look at the purpose and scope of this Special Issue and provide a clear justification of the relevance of the investigation for improving healthcare (internationally) and provide 3 -4 recommendations based upon the finding of their research that may lead to the enhancement in the quality and delivery of healthcare.

Author Response

Reviewer 2

This is an interesting manuscript that addresses an area of importance. There are several factors that need to be addressed.

Thank you for your compliments and your well-considered comments. We hope we sufficiently clarify your points in the answers and article.

Q1.1 Please can the authors provide an account of what ethical approvals were sought and how they gained the necessary permissions to access the healthcare professionals.

A1.1 Permission to access the professionals was given by the director of the nursing home organization.

Q1.2 How as consent obtained?

A1.2 Oral consent was obtained at the beginning of each focus group. (in article in line 117 page 3)

Q1.3 What information was provided to potential participants.

A1.3 The researcher approached one member of the multidisciplinary team in all nine nursing homes of the organization by email. In this email, the goal of the focus group was explained, as the duration and inclusion criteria. Also, this member was asked in the email if they could ask if several multidisciplinary team members could participate in the focus group at their location. The member of the team who was approached by the researcher, contacted the multidisciplinary team members also by email or face-to-face to see if they could participate in the focus group. We’ve clarified this process in more detail in the article (line 95, page 3).

Q1.4 How did the researchers ensure anonymity and confidentiality?

Q1.4 Anonymity and confidentiality were guaranteed, the names of people, appearing in the quotes, were deleted from the written texts or were anonymized. (this has been added in line 142 page 4)

Q1.5 Where were the focus groups conducted?

A1.5 The focus groups were conducted in a meeting room in the nursing home where the participants work.

Q1.6 What was the role of the Occupational Therapy student in the investigation and who invited them to join the research team?

A1.6 The occupational therapy student was asked by the researchers to be a part of the study and had no other obligations in the organization. She moderated two of the focus groups under the supervision of author 1. She had a role in the analysis of the data as she made transcriptions and coded the data with author 1. Also, she helped with reviewing and editing the article (see article line 129 page 2 and line 409 page 10).

Q2.1 Recruitment of the participants was via e-mail (page 2 line –75) how did the researchers obtain this personal data and gain access to this information? What information was sent via e-mail?

A2.1 Permission to access the professionals was given by the director of the nursing home organisation. The researcher approached one member of the multidisciplinary team in all nine nursing homes of the organization by email. In this email, the goal of the focus group was explained, as the duration and inclusion criteria. Also, this member was asked in the email if they could ask if several members of the multidisciplinary team could participate in the focus group. The member of the team who was approached by the researcher contacted the multidisciplinary team members also by email or face-to-face. We’ve clarified this process in more detail in the article (line 95, page 3).

Q2.2 Did the researchers work in the participating institutions?

A2.2 The first author works as a lecturer practitioner in the nursing home organization which participated in the study, is a Ph.D. student at LUMC PHEG and has eight years of experience as a science practitioner by The Hague University of Applied Sciences. Characteristics are reported about author 1 in the article line 119 page 3  as followed: The researcher is an experienced group facilitator, is employed in the organization as a lecturer practitioner and has worked with multidisciplinary teams to maintain meaningful activities. The second author was a student with no other obligations in the organization. The third author works as an occupational therapist in the nursing home organization which participated, although she was not working in the participating institutions in the multidisciplinary teams. Author four and five did not work in the participating organization.

In the discussion (line 384 page 10) we added a limitation: Another limitation was that was that authors one, who moderated the focus groups also worked in the organization. Although we cannot rule out that this had an effect on the participants responses in the focus groups [1], we do not have any indication for different responses in the focus groups because of the affiliation of the researchers. Besides, the cooperation with researchers that are well experienced in Dutch nursing home care, but outside this organization reduced the risk of overly biased results.  

Q3.1  Page 2 line 89 – 90  “The topic list for the main subject was derived from the experiences of the researcher (author 1) and an occupational therapist (author 3). What did the topic list comprise of?

A3.1 The topic list can be found in table 1 in the article (line 121, page 3) and comprises:

Head topic

Sub topic

What is a meaningful activity?

Can you give an example of a meaningful activity?

What do you believe makes an activity meaningful?

How are meaningful activities assessed?

Opinions and current practice:

What do you consider barriers and facilitators?

How do you collaborate as a multidisciplinary team in the assessment of meaningful activities?

How do you maintain meaningful activities during the transition from home to nursing home?

Opinions and current practice:

What do you consider barriers and facilitators?

How do you collaborate as a multidisciplinary team in maintaining meaningful activities?

Q3.2 What is the experience of the researcher?

A3.2 The first author, who conducted the focus groups, has experience in action research and practice development and has several years of experience in collecting qualitative data. She works as a lecturer practitioner in the nursing home organization which participated in the study, is a Ph.D. student at LUMC PHEG and has eight years of experience as a science practitioner by The Hague University of Applied Sciences. The second author, who also conducted the focus groups with author 1, had no experience in collecting qualitative data. She has a bachelor’s degree in Psychology and at the time of the research, she was an occupational therapy student in her last year. The third author has 13 years of experience as an occupational therapist in the care for persons with dementia and she is specialized in learning abilities for persons with dementia. The fourth author is a professor of institutional care and elderly care medicine at Leiden University Medical Center. He is also an elderly care physician and medical director of Topaz (Leiden) and a visiting professor at Bergen University (Norway). He is an expert in quantitative and qualitative research. The fifth author is an assistant professor at LUMC. She is very experienced in quantitative and qualitative research.

Q4 Can an overview of the 22 participants be presented – this is briefly outlined on page 4 lines 121 -129. Was any further demographic information collected – if so, can a summary be provided? It would be useful to provide participant information for each of the focus groups.

A4 We presented a table below and added a summary of the characteristics in the article line 155 page 4 In total four men and 18 women participated. Work experiences in the care for persons with dementia varied: four participants had 0-2  years’ experience, nine participants 2-10 years and seven participants more than 10 years, from two participants we had no data. Six participants were between 20-35 years of age, nine participants between 35-50, three participants between 50-65, from four participants we had no data.

Q5.1 The term fragment is used page 3 lines 100 -117 – please can you clarify are you referring to extracts/excerpts of the data transcripts –

A5.1 Yes, the fragments were extracts of the data transcripts

Q 5.2 also who undertook the transcription of the focus group interviews?

A5.2 As noted in line 127 page 3 author 2 first transcribed the focus groups. Author 1 listened to the audio recording and also read the transcripts.

Q6 Please can the authors look at the purpose and scope of this Special Issue and provide a clear justification of the relevance of the investigation for improving healthcare (internationally) and provide 3 -4 recommendations based upon the finding of their research that may lead to the enhancement in the quality and delivery of healthcare.

A6. In the introduction, we have added a recent study which has been published about the adjustment period and the perspectives of persons with dementia, family members and facility staff [2]. We have added the results of this study because this study, highlights the relevance of the investigation. The sentences which were added are: A recent study investigated adjusting to residential aged care facilities from the perspectives of persons with dementia, family members and facility care staff and identified meaningful activities as critical for facilitating adjustment to life in the facility (line 46 page 2). Also the study of Davison et al. [2] found that  although some residents continued with previous hobbies, many residents and families found activities inadequate to meet residents’ needs for stimulation and interest. In this study staff found it difficult to find activities that meets individual interests, preferences and capacities of persons with dementia and to incorporate meaningful activities in day-to-day-life (line 61 page 2).

To our best knowledge, this explorative study is the first study that focus completely on maintaining meaningful activities for persons with dementia in the transition period from home to a nursing home from the perspective of health care professionals.  (in article line 391 page 10)

In the search for healthcare adjusted to the need of older people and to support older people to live the life they wish to live, person centred care, this study provides valuable information. (last part of sentence in article line 397 pag 10).  Especially because the transition is experienced as a major life event for persons with dementia and their family members [3] and adjustment is difficult [4]. Meaningful activities are mentioned as essential for adjusting to residential care [2].  A range of factors are associated with activity involvement in the nursing home and relate to the person’s interests, competences, abilities, the (social and physical) environment and other factors related to the organization of care [5].  (above sentences are present in the introduction) Providing personal meaningful activities for persons with dementia is a challenge for health care professionals and health care organisations [4,6,7]. Furthermore, there is a lack of guidelines [2] and interventions [4,8] focusing on the transition from home to a nursing home. (sentences are added in line 393 page 10). The findings of this study provide valuable insights, both for professionals and organizations, on how to maintain meaningful activities during this transition. (in article line 396 page 10).

What this paper adds:

  • This study shows that health care professionals experienced a lack of attention and awareness among health care professionals within their nursing home organizations which could have an influence on the identification and maintenance of meaningful activities in transition from home to a nursing home (added in article in line 358 till 361 page 9).
  • Paying specific attention to collecting information about meaningful activities before and after the move to the nursing home and adapting meaningful activities to the new situation can contribute to the maintenance of meaningful activities (in the article line 355 page 9 , the word maintaining was added)
  • Improving transitional care focused on maintaining meaningful activities, an interdisciplinary approach, both with professionals in the nursing home and professionals involved prior to the admission, is therefore recommended. (in article line 372 page 10)

Several recommendations could be made for health care professionals and organizations based on the findings in the  present study:

  • Training of healthcare professionals can contribute to awareness among healthcare professionals of the importance of meaningful activities in the transition period, knowledge of the factors influencing the maintenance of meaningful activities, risks of losing meaningful activities and possibilities to maintain meaningful activities. (added in article in line 399 page 10)
  • An interdisciplinary approach, both with professionals in the nursing home and professionals involved prior to the admission, is recommended. (in article line 372 page 10)
  • The development of an interdisciplinary multicomponent intervention and guidelines aimed at supporting persons with dementia and their families during this important transitional period (in article line 402 page 10)

Recommendations for further research

  • Further research should focus on how interventions help persons with dementia and family members to maintain their lives as fully as possible during the transitional period and how activities can be adapted to remain meaningful for them, despite progression of the disease and changing environments (in article line 405 page 10)

Table 1. Participants demographics

Focus group

n

occupation

gender

Work experience in the care for persons with dementia

age

1.

6

Occupational therapist (n=2), Activity therapist (n=1),Team manager (n=1), Nurse aides (n=1), Psychologist (n=1)

Five women

one men

2-10 years: n=1

>10 years: n=3

No data: n=2

35-50 year: n= 3

50-65 years: n=1

No data: n=2

2.

3

Physician(n=1),Elderly care physician (n=1), Psychologist(n=1)

One women

Two men

0-2 years: n=1

2-10 years: n=1

>10 years: n=1

20-35: n=1

35-50 year: N=2

3.

6

Occupational therapist(n=1), Psychologist(n=2), nurse aides(n=1), Physiotherapist(n=2)

Six women

0-2 years: n=1

2-10 years: n=4

>10 years: n=1

20-35: n=3

35-50 year: N=2

No data: n=1

4.

7

Psychologist (n=1), nurse aide(n=1), registered nurse(n=1), hostess(n=1), team manager(n=1), physiotherapist(n=2)

Six women

One men

0-2 years: n=2

2-10 years: n=3

>10 years: n=2

20-35: n=2

35-50 year: N=2

50-65 years: n=2 No data: n=1

total

N=22

five psychologists, four physiotherapists, three nurse aides, one registered nurse, one hostess (a paid assistant without nurse training), one activity therapist, three occupational therapists, two team managers, one physician and one elderly care physician.

18 women

4 men

0-2 years: n=4

2-10 years: n=9

>10 years: n=7

No data: n=2

20-35: n=6

35-50 year: n= 9

50-65 years: n=3

No data: n=4

References

  1. Tong, A.; Sainsbury, P.; Craig, J. Consolidated criteria for reporting qualitative research (COREQ): a 32-item checklist for interviews and focus groups. Int J Qual Health Care 2007, 19, 349-357, doi:10.1093/intqhc/mzm042.
  2. Davison, T.E.; Camões Costa, V.; Clark, A. Adjusting to life in a residential aged care facility: Perspectives of people with dementia, family members and facility care staff. Journal of Clinical Nursing 2019, 10.1111/jocn.14978, doi:10.1111/jocn.14978.
  3. Afram, B.; Verbeek, H.; Bleijlevens, M.H.; Hamers, J.P. Needs of informal caregivers during transition from home towards institutional care in dementia: a systematic review of qualitative studies. Int Psychogeriatr 2015, 27, 891-902, doi:10.1017/S1041610214002154.
  4. Sury, L.; Burns, K.; Brodaty, H. Moving in: adjustment of people living with dementia going into a nursing home and their families. Int Psychogeriatr 2013, 25, 867-876, doi:10.1017/S1041610213000057.
  5. Smit, D.; de Lange, J.; Willemse, B.; Pot, A.M. Predictors of activity involvement in dementia care homes: a cross-sectional study. BMC Geriatr 2017, 17, 175, doi:10.1186/s12877-017-0564-7.
  6. Tak, S.H.; Kedia, S.; Tongumpun, T.M.; Hong, S.H. Activity Engagement: Perspectives from Nursing Home Residents with Dementia. Educ Gerontol 2015, 41, 182-192, doi:10.1080/03601277.2014.937217.
  7. den Ouden, M.; Bleijlevens, M.H.; Meijers, J.M.; Zwakhalen, S.M.; Braun, S.M.; Tan, F.E.; Hamers, J.P. Daily (In)Activities of Nursing Home Residents in Their Wards: An Observation Study. J Am Med Dir Assoc 2015, 16, 963-968, doi:10.1016/j.jamda.2015.05.016.
  8. Muller, C.; Lautenschlager, S.; Meyer, G.; Stephan, A. Interventions to support people with dementia and their caregivers during the transition from home care to nursing home care: A systematic review. Int J Nurs Stud 2017, 71, 139-152, doi:10.1016/j.ijnurstu.2017.03.013.

Reviewer 3 Report

Nicely done and well written

Author Response

We thank the reviewer, for their compliments and time spent on reviewing the article.